# Preparation, Structural Characterization and Evaluation of Some Dynamic and Rheological Properties of a New Type of Clay Containing Mastic Material, Clay-Mastic

Ahmet Gürses [1],* and Tahsin Barkın Barın [2]

1   Department of Chemistry Education, K.K. Education Faculty, Atatürk University, Erzurum 25240, Turkey
2   Nanoscience and Nano Engineering Department, Atatürk University, Erzurum 25240, Turkey
*   Correspondence: ahmetgu@yahoo.com or agurses@atauni.edu.tr

**Abstract:** This study focused on the preparation, structural characterization and evaluation of some dynamic and rheological properties of a new type of mastic material, clay-mastic, which consists of bituminous binders mixed with mineral fillers. For this purpose, mastic samples were prepared by mixing conventional bitumen (50/70) with organo-montmorillonite (OMMT) in various proportions. X-ray powder diffraction (XRD) spectra and scanning electron microscope (SEM) and high-resolution transmission electron microscope (HRTEM) images of raw clay (MMT), organo-clay (OMMT) and raw bitumen with the prepared mastics were taken, and the changes in the crystallographic properties of the clay and its dispersion characteristics in the bitumen matrix as well as the changes in the morphological properties of the mastic samples were investigated comparatively. In addition, penetration, softening point, flash point, dynamic viscosity, dynamic shear rheometer (DSR) and Fraass breaking point tests were carried out together with those of base bitumen in order to evaluate the properties of the prepared mastic samples in terms of dynamics and rheology. A comparison of the images of raw clay and organo-clay indicated delamination based on surface modification in clay layers in those belonging to organo-clay, and diffractograms of prepared mastic samples showed that the characteristic smectite peak of Montmorillonite shifted to the left gradually with an increasing clay ratio. This shows that due to the successful lyophilic modification on the clay surface, the effective intercalated and even exfoliated dispersion of the clay layers in the bitumen matrix can occur. The penetration viscosity number (PVN) values, defined as a function of penetration and dynamic viscosity, and the penetration index (PI) values, defined as a function of penetration and softening point, were found to be within a well-accepted thermal stability range for all of the prepared mastic samples. For this reason, it was concluded that the sensitivity of the samples to temperature decreased with the addition of organo-clay, thus providing applicability in a wider temperature range. The Fraass breaking point and dynamic viscosity values of the prepared mastic samples decreased and increased, respectively, with an increasing clay ratio, meaning that the addition of organo-clay lead to an increase in the crack resistance of the samples at low temperatures and a decrease in their permeability.

**Keywords:** mastic material; organo-clay; bitumen; modification; dynamic and rheological properties





## 1. Introduction

Mastics are materials used especially in asphalt mixtures, consisting of a combination of fillers and bituminous binders, and the characteristics of a mastic can affect the performance of the final asphalt mixtures and many properties such as stiffening, workability, fatigue cracking and moisture sensitivity [1]. The properties of mastic materials depend on many factors, such as the binder and filler properties and filler–binder ratio. Additionally, since there are many types of filler and binders available, and a particular type of binder may exhibit different adhesion characteristics with different fillers, the performances of

mastic materials are highly dependent on the adhesion properties of the filler and binder. Fillers are usually found in aggregates as the fine powder fraction resulting from the crushing of aggregates in different fractions. Besides the natural powders sieved from the collected crushed aggregates, the most common type of mineral filler added to the mixtures is limestone, but fillers such as slaked lime, fly ash or slag and Portland cement can also be used [2]. It is claimed that the filler has two main functions: to serve as a volume-filling material in mixtures, in the cavities of coarser aggregates and to act as an extender of the binder as an active material that adsorbs the components in the binder. Density is another essential property of fillers, and the amount of filler used in asphalt mixtures depends on the mass ratio of the components. When the density and specific gravity of different fillers change, the volume ratio of filler added to asphalt mastics similarly changes. The filler density is a function of various physical factors such as particle geometry, particle size distribution and morphology. The binder is present in asphalt mastics in a free and fixed state, and the fixed binder can be divided into two parts: the part that adheres to the filler material and acts as a part of the particle and the part that is not adsorbed but is affected by the adsorption of the interior [3]. Adhesion occurs by the interaction of the fixed binder and the filler, and the strength of the adhesion depends on the surface activity and mineralogy of the filler. It has been found that a higher geometric irregularity of the filler results in a higher adsorption density. This may result in the strengthening of the binder–filler interaction and a relative increase in the amount of fixed binder, resulting in a higher consistency and strength of the mastic. The clay content in fillers is critical, and clay has several properties that are critical for asphalt mixes [4]. The first is the relatively large surface area of clay minerals, which require more bitumen for pavement, and the other is that clay minerals can become plastic when exposed to water. Montmorillonite is generally considered harmful because of its tendency to retain moisture, an undesirable behavior during freeze-thaw cycles [4]. High moisture content in the filler weakens the filler–binder interaction in the mastic, and water absorption reduces the frost resistance of the asphalt mixture. Moreover, clays and friable particles have a tendency to form agglomerates that can break apart under load or stress. A low amount of clay that can expand and swell in dense areas can impart uneven strength and have critical effects on mastic properties [5]. Environmentally friendly natural montmorillonite is primarily used to develop modified bitumens containing styrene-butadiene-styrene triblock copolymer, poly (styrene b-ethylene-co-propylene), poly phosphoric acid and poly (ethylene terephthalate). It can improve the hardness, rutting and moisture resistance and storage stability of bitumen while reducing its low temperature fluidity and ductility. Moreover, montmorillonite can significantly improve the adhesion strength of bitumen to siliceous substrate under shear by adsorbing asphaltenes and other bitumen components and acids. Since a particular type of binder can interact differently with different fillers, the properties of the mastic depend on both the mastic and the binder and their combination. Additionally, the effect of the filler properties on the mastic can vary with the type of binder and the gradation of coarse aggregates [5]. The relative viscosity of the binder has the greatest effect on the rutting potential, while the filler-to-binder ratio has a significant effect on the mastic performance. Due to differences in specific gravity and density, filler materials can have a predetermined weight relationship of filler and binder and large volume differences depending on the type of filler used in the mastic. If the filler density is low, the filler volume increases for a given amount of filler by weight, and more binder is required to cover the filler material. This can increase the relative amount of fixed binder, resulting in less free binder and dry asphalt mastic and an insufficient coating of aggregate particles [3]. The superpave specifications recommend a filler-to-binder ratio of 0.6 to 1.2 by mass for asphalt mixes, where the binder content is defined as the free binder content that is not absorbed by the aggregates in the mix. The filler acts as a binder reinforcement, and the reinforcement mechanism can be considered based on particle interaction and volume filling [1]. With increasing filler content, particle interaction reinforcement increases as the filler material gets closer and forms a skeleton. The stiffening effect of the volume-filling is a result of harder filler

materials in a less rigid mastic, which makes it denser. The physiochemical reinforcement stiffening effect results from interactions at the interface between the asphalt binder and the filler, i.e., adsorption [1]. The main factors affecting the physicochemical properties are mineral composition, surface texture, surface activity and other structural properties. Adsorption is exothermic, and heat is released by the filler–bitumen interaction. Both high geometric irregularity and high surface activity strengthen the filler–binder interaction, and the ratio of fixed binder to free binder also increases. This physicochemical effect results in better consistency mastic and higher-strength asphalt mixtures. It has been reported that limestone and hydrated lime exhibit the highest selective adsorption to the binder, followed by basalt, while sandstone shows almost no adsorption, and siliceous fillers also show low adsorption due to their low surface activity. The selective adsorption capacity determines the effect of the filler on the adhesion and stability behavior of the asphalt mixture [6]. Another mastic property that is influenced by filler type is irrecoverable creep compliance, which indicates permanent deformation of the mastic during repeated loading.

Mastic material, which is a mixture of bitumen and very fine filler particles, can significantly affect not only the rheological properties of the treated asphalt mixture but also the viscoelastic properties of the compacted asphalt layers, including resistance to crack formation. However, the number of scientific studies that deal with this issue in detail is quite limited. The aim of this study is to determine the basic rheological properties of these mastic materials using a dynamic shear rheometer, focusing specifically on the petrographic origin of the filler particles.

The physical and chemical properties, ingredients and proportions of the input materials have a great influence on the mechanical properties of asphalt mixtures used in road construction [7–23]. While preparing these mixtures, the bitumen is mixed with the aggregate fractions to be covered by the bitumen. A significant part of the aggregate is filler, which is characterized by the predominant content of fine particles smaller than 0.063 mm in size and primarily serves to stiffen the binder. The filler, which has the largest specific surface area among very fine grained and all aggregate fractions, binds a large amount of bituminous binder and thus forms a mastic mortar consisting of a mixture of filler and bituminous binder in the asphalt mixture, which greatly affects the properties of asphalt mixtures. Depending on the mineralogical properties of the aggregate used during the compaction (rolling) of the asphalt mixture, adverse effects may occur on the rheological properties of the asphalt mixture. The problems are generally thought to be caused by the presence and transfer of phyllosilicate particles, which are relatively soft but have a very high bulk density and can be characterized by very good basal separation, that come out of the bedrock during the crushed aggregate production process [24–27].

Bitumen, which is fluid at high temperatures and brittle at low temperatures, is suitable for use in a limited temperature range. Therefore, bitumen, which is a very temperature sensitive binder material, can be the source of negative results such as rutting or cracking in asphalt pavements. Various modifications have been made to improve the mechanical and thermal behavior of bitumen [28]. Although bitumen modification with polymer additives is widely used and accepted by many researchers as an effective way to improve bitumen behavior, polymer-based modification also has many disadvantages such as the increased hardness of bitumen, deterioration of low temperature properties and high temperature requirement for homogeneous dispersion. Nanomaterials such as silica, ceramics, clay, oxides and inorganic particles, as well as organic and functionalized nanoparticles, have the ability to reduce the incompatibility between some natural aggregates and bitumen binders and provide more sustainable and durable coating solutions, exhibit antioxidant effects or reduce photo-induced aging and asphalt flammability. They can also act as an anti-peeling agent or promote adhesion between bitumen and acid aggregates such as granite [28–31]. Therefore, studies on the use of different organic and inorganic alternative additives have gradually increased [32]. Particularly, organo-clay-reinforced polymer modified bitumen mixtures have also been prepared to improve the polymer-bitumen interface properties. More intense interactions between polymers and bitumen

can affect the microstructure of the polymer–bitumen dispersion, and with the addition of organo-clay, the interaction between polymer and bitumen can be improved, resulting in a more homogeneous dispersion [33]. By modifying the bitumen using nanocomposites, new-generation asphalt mixtures with improved performance, stability and mechanical properties can be obtained [34,35]. In asphalt mixtures modified with organo-clay and carbon microfiber, significant improvements were achieved in mechanical properties and water resistance [36,37]. Additives added to bitumen vary widely, but organo-clay and fiber applications are particularly important. It was observed that the addition of nanoclay and micro carbon fiber to the bitumen resulted in a significant decrease in the water sensitivity of the asphalt mixture while also reducing the penetration and increasing the softening point and viscosity [36,38,39]. In addition, it was found that the tensile strength, creep and fatigue resistance of bitumen modified with cloisite organo-clay were increased [40]. Thus, it can be argued that organo-clay may be a serious alternative for effective bitumen modification [41–43].

This study aims to prepare and structurally characterize a new type of mastic material, clay mastic, which is combination of organo-clay and bituminous binder, as well as to investigate the changes in some dynamic and rheological properties of the mastic with the filler/binder (organo-clay/bitumen) ratio.

## 2. Materials and Methods

### 2.1. The Preparation of Organo-Clay

Super organophilic clay to be used as filler was prepared by solution intercalation method using a cationic surfactant, Cetyltrimethyl ammonium Bromide, CTAB (purchased from Merck Inc.), long chain hydrocarbon material (Table 1) and montmorillonite clay from Çankırı region in Turkey (Table 2). For this purpose, by adding cationic surfactant (CTAB) that can meet the concentration of 160 mg/L to the aqueous mixture, the dispersion of the long-chain hydrocarbon substance added at the ratio of 0.3 g/1.0 g in water was achieved, and then the dispersion was mixed mechanically for 30 min at a mixing speed of 50 min$^{-1}$. Then, a certain amount of raw clay was added to the hydrocarbon-water dispersion, and mixing was continued for 30 min at a mixing speed of 200 min$^{-1}$. Finally, the resulting mixture was filtered and dried in a vacuum oven at 110 °C for 2 h, then ground, passed through a 200-mesh sieve in ASTM standard and stored in a closed container for future experiments [44]. The distance between plates, which was 1.23 nm for raw clay, increased to 1.66 nm for organo-clay. The zeta potential of raw clay, which was −34.0 mV, decreased to 0.0 mV after hydrophobization. The density of organo-clay with a static contact angle of 107° is also 1.95 g/cm$^3$.

**Table 1.** Some characteristics of long-chain hydrocarbon material.

| Density (15 °C), kg/m$^3$ | Calorific Value MJ/kg | Flash Point °C | Water by Distillation, wt. % | C | H | N | S | Ash |
|---|---|---|---|---|---|---|---|---|
| 990.7 | 42.74 | 105.8 | 0.1 | 83.4 | 11.9 | 0.8 | 1.5 | 0.03 |

**Table 2.** Mineralogical content of montmorillonite.

| Components (%) | | | | | | | | | |
|---|---|---|---|---|---|---|---|---|---|
| SiO$_2$ | Al$_2$O$_3$ | Fe$_2$O$_3$ | MgO | CaO | Na$_2$O | K$_2$O | TiO$_3$ | SO$_3$ | Other |
| 59.32 | 17.19 | 5.95 | 3.63 | 2.21 | 1.68 | 0.97 | 0.74 | 0.51 | 7.81 |

### 2.2. Preparation of Organoclay-Bituminous Mastic Material

The base bitumen with a penetration grade of 50/70 used in this study was obtained from TÜPRAŞ Inc. İzmir, Turkey. This normal road pavement bitumen was chosen with the thought that after the addition of organo-clay, the bitumen would improve the rutting and

fatigue strength performance in hot regions and be resistant to cracking in cold climatic regions. Some basic properties of the bitumen used were determined by tests such as penetration, softening point, flash point, elastic recovery, dynamic viscosity, dynamic shear rheometer and Fraass breaking point, and the obtained values are given in Table 3 together with the relevant standards.

**Table 3.** Some properties of the base bitumen.

| Test | Value | Standard |
|------|-------|----------|
| Penetration (25 °C, 100 g, 5 min) | 54.6 | TS 1081 EN 12591 |
| Softening Point (°C) | 49.2 | TS EN 1427, ASTM D36 |
| Flash Point (°C) | 342 | TS EN ISO 2592, TS 1171 |
| Elastic Recovery % | 0 | TS EN 13398 |
| Dynamic Viscosity (Cp) | 440 | TS 1451 EN ISO 3104 |
| DSR (Dynamic Shear Rheometer) (GRADE) | 64 | TS EN 14770, AASHTO T315 |
| Fraass Breaking Point (°C) | −7 | TS EN 12593 |

Organo-clay-bituminous mastic mixtures were prepared by mixing in a mechanical mixer at 110 °C at a mixing speed of 150 $min^{-1}$ for 1 h. Basic rheological tests such as penetration (ASTM D5), ductility (ASTM D113) and softening point (ASTM D36) were applied to the prepared mastic mixtures. Additionally, the relevant test results of different polymer modified bitumen mixtures (PMB) are presented in Table 4 to compare with the results of the prepared mastic mixtures [45].

**Table 4.** Test results obtained for various polymer modified bitumen mixtures (PMB).

| Test | PMB 64-28 | PMB 70-16 | PMB 70-22 | PMB 70-28 | PMB 76-16 | PMB 76-22 |
|------|-----------|-----------|-----------|-----------|-----------|-----------|
| Penetration (25 °C, 100 g, 5 min) | 50–90 | 30–70 | 30–90 | 20–60 | 20–70 | 10–50 |
| Softening Point (°C) | 52 | 62 | 62 | 62 | 67 | 67 |
| Elastic Recovery % | 80 | 60 | 70 | 80 | 60 | 70 |
| Flash Point (°C) | 220 | 220 | 220 | 220 | 220 | 220 |
| DSR (Dynamics Shear Rheometer) (GRADE) (G/Sin δ > 1 kPa) | 64.0 | 70.0 | 70.0 | 70.0 | 76.0 | 76.0 |

The codes of the mastic mixtures prepared in different filler/binder (organo-clay/bitumen) ratios and the corresponding ratios are given in Table 5.

**Table 5.** Codes and corresponding ratios of mastic mixtures prepared in different filler/binder (organo-clay/bitumen) ratios.

| Specimen Code | Type of Bitumen | Filler | Filler/Binder Ratio |
|---------------|-----------------|--------|---------------------|
| B | 50/70 | Base Bitumen | 0.00 |
| CM1 | 50/70 | Hydrocarbon-doped organo-clay | 0.02 |
| CM2 | 50/70 | Hydrocarbon-doped organo-clay | 0.04 |
| CM3 | 50/70 | Hydrocarbon-doped organo-clay | 0.06 |
| CM4 | 50/70 | Hydrocarbon-doped organo-clay | 0.08 |
| CM5 | 50/70 | Hydrocarbon-doped organo-clay | 0.10 |

2.2.1. The Penetration Index (PI)

Penetration index used to determine the heat sensitivity of bituminous binders is calculated as a function of penetration and softening point. The PI value is calculated using the formula given in Equation (1), and its optimal value is suggested to be between −2 and

+2. It is suggested that if bitumen is greater than +2, it may oxidize, and if it is less than −2, its sensitivity to temperature will increase excessively [46].

$$PI = \frac{1952 - (500 \times \log P_{25}) - (20 \times T_{YN})}{(50 \times \log P_{25}) - T_{YN} - 120} \tag{1}$$

where $P_{25}$ is the penetration value of bitumen at 25 °C, and $T_{YN}$ is the softening point at 135 °C.

### 2.2.2. Penetration Viscosity Number (PVN)

Penetration viscosity number (PVN), which is used as a measure of temperature sensitivity based on the penetration index, is considered to have a high temperature sensitivity of bitumen, similar to the penetration index value if the PVN value is low. Generally, PVN value between −1 and +1 is considered optimal. The PVN value can be calculated using the formula given in Equation (2) [47].

$$PVN = -1.5 \times \frac{4.258 - (0.7967 \times \log P_{25}) - \log(V)}{0.7951 - (0.1858 \times \log P_{25})} \tag{2}$$

where V is the kinematic viscosity (centistokes) at 135 °C, and $P_{25}$ is the penetration at 25 °C.

### 2.3. High Resolution Transmittance Electron Microscopy (HRTEM) Analysis

HRTEM analysis is of great importance in examining the textural structures of materials at nanoscale and especially the dispersions of nanoparticles in nanocomposites [48]. In this study, HRTEM images of both raw clay and organo-clay were taken in order to characterize the textural structures and to monitor the degree of lyophilization. For this, HRTEM images of both raw montmorillonite and organo-clay were taken using the JEOL JEM-2100F high-resolution transmittance electron microscope (LaB6 filament) (HRTEM) operating at 200 kV.

### 2.4. Scanning Electron Microscopy (SEM) Analysis

In order to observe and evaluate the morphological changes that may occur in mastic mixtures due to adhesion interactions between organo-clay and bitumen, scanning electron microscope (SEM) images of both base bitumen and organo-clay-bitumen mastic mixtures prepared in various ratios were taken at 30 kV using a scanning electron microscope (SEM) (FEI-INSPECT S50 model).

### 2.5. X-ray Powder Diffraction Spectroscopy (XRD) Analysis

In layered clays such as montmorillonite, the variation of the interlayer spacing can be considered to determine the type of composite prepared [49,50]. X-ray powder diffraction (XRD) spectra were taken to examine the behavior of bitumen in the interlayer region and deformations in the crystal structures in organo-clay-bitumen mastic mixtures. XRD diffractograms of organo-clay, base bitumen and organo-clay-bituminous mastic mixtures prepared in various proportions were taken using Rigaku brand Smart Lab model XRD device in 2θ = 2–40° scanning area and 2 °/min scanning speed with a Cu Ka (λ: 1.5404) radiation source.

## 3. Results and Discussion

### 3.1. High Resolution Transmission Electron Microscope (HRTEM) Analysis of Organo-Clay (OMMT)

In order to compare the differences in the textural structures of raw montmorillonite (MMT) and organo-montmorillonite and to see the effectiveness of the modification made for lyophilization, HRTEM images of both were taken and are given in Figure 1a,b, respec-

tively. The long, dark lines seen in the HRTEM images correspond to grains with a layered structure.

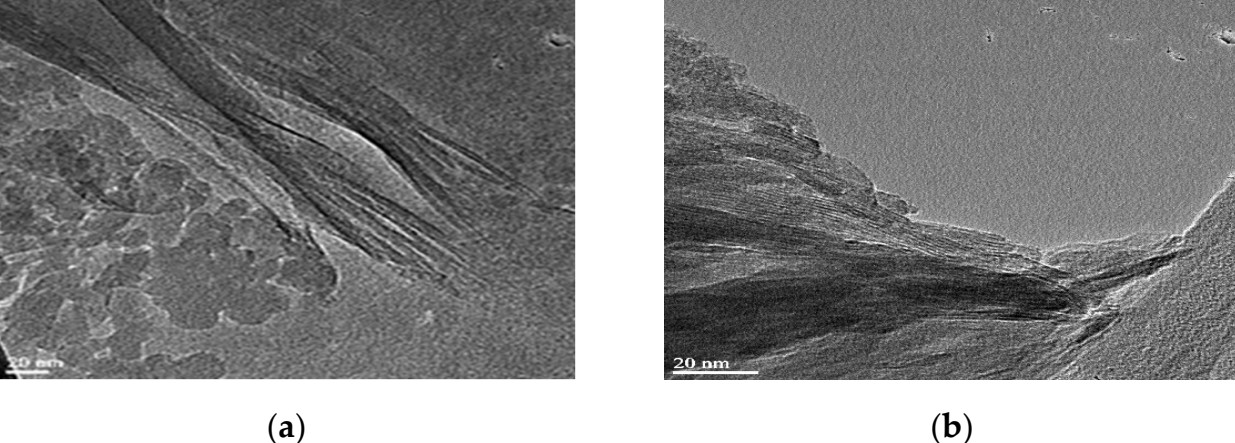

(**a**)

(**b**)

**Figure 1.** HRTEM images of raw montmorillonite (MMT) (**a**) and organo-montmorillonite (OMMT) (**b**).

As can be seen from Figure 1b, in the case of organo-clay, the clay layers appearing as aggregates of long fibers appeared with wider interlayer distances compared to raw clay. Accordingly, it can be argued that due to the modification, organo-clay particles with lyophilic interfaces and more spaces become more suitable for both bitumen penetration and adhesion interactions.

*3.2. X-ray Powder Diffraction (XRD Analysis of Mastic Mixtures Prepared at Different Filler/Binder (Organo-Clay/Bitumen) Ratios*

XRD diffractograms of mastic mixtures (CM1–6) prepared at different filler/binder (organo-clay/bitumen) ratios, together with the diffractograms of raw montmorillonite, organo-montmorillonite and base bitumen (B), are given in Figure 2. From Figure 2, it can be seen that the characteristic smectite peak of montmorillonite with a 2.1-layered structure gradually expanded and shifted to the left in organo-montmorillonite. The distance between plates, which was 1.23 nm for raw clay, increased to 1.66 nm for organo-clay. The smectite peak observed at 2 $\theta$:6.2 in raw clay shifted to 5.2, which may indicate bitumen intercalation in the interlayer space. On the other hand, calcite, quartz and montmorillonite peaks appeared at 28.9 and 2 $\theta$ values corresponding to 26.0 and 21.2, respectively, and the peak at 21.2 was deformed in organo-clay and mastic mixtures. This clearly indicates the expansion in interlayer spacing. The diffractograms of mastic mixtures prepared at different filler/binder ratios also show that the intensity of the smectite peak gradually increases with increases in the organo-clay ratio, but the smectite peak disappears almost completely, especially at low organo-clay ratios, and the others are partially shifted to the left and have relatively lower peak intensity compared to organo-clay.

The variation observed at low organo-clay ratios indicated that the clay plates were well-dispersed in the bitumen matrix. In other words, such deformation of the smectite peak means that the clay layers exfoliated and the interlayer space expended too much [49]. It can also be argued that exfoliated clay layers increased the interaction surface of the bitumen particles and improved their displacement ability. However, the partial aggregation of the clay layers, which may occur at high clay ratios, may also have the potential to limit the freedom of displacement of the bitumen particles.

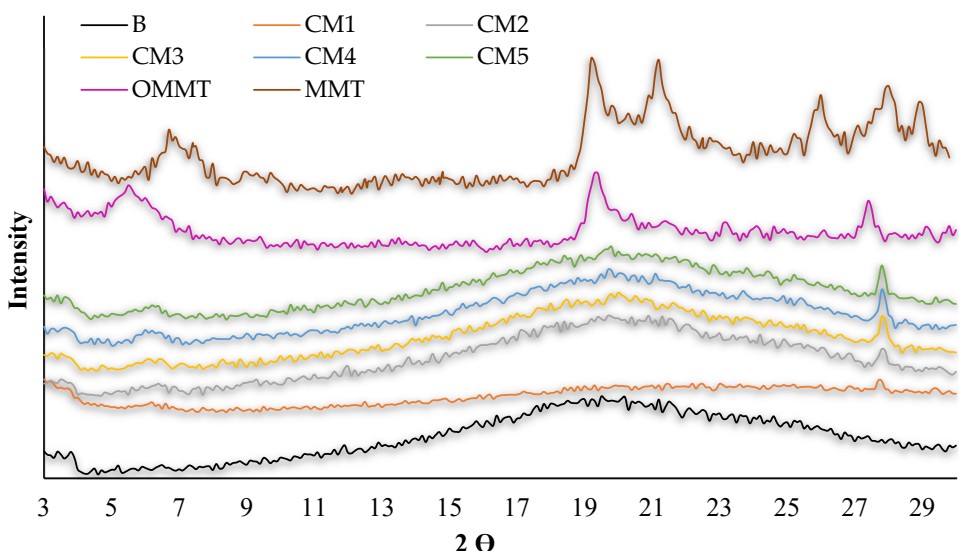

**Figure 2.** XRD diffractograms of mastic mixtures (CM1–5) prepared at different filler/binder (organo-clay/bitumen) ratios, raw montmorillonite (MMT), organo-montmorillonite(OMMT), and base bitumen (B).

*3.3. Scanning Electron Microscopy (SEM) Analysis of Mastic Mixtures Prepared in Different Filler/Binder (Organo-Clay/Bitumen) Ratios*

SEM images of mastic mixtures (CM1-5) prepared in different filler/binder (organo-clay/bitumen) ratios and base bitumen (B) are given in Figure 3.

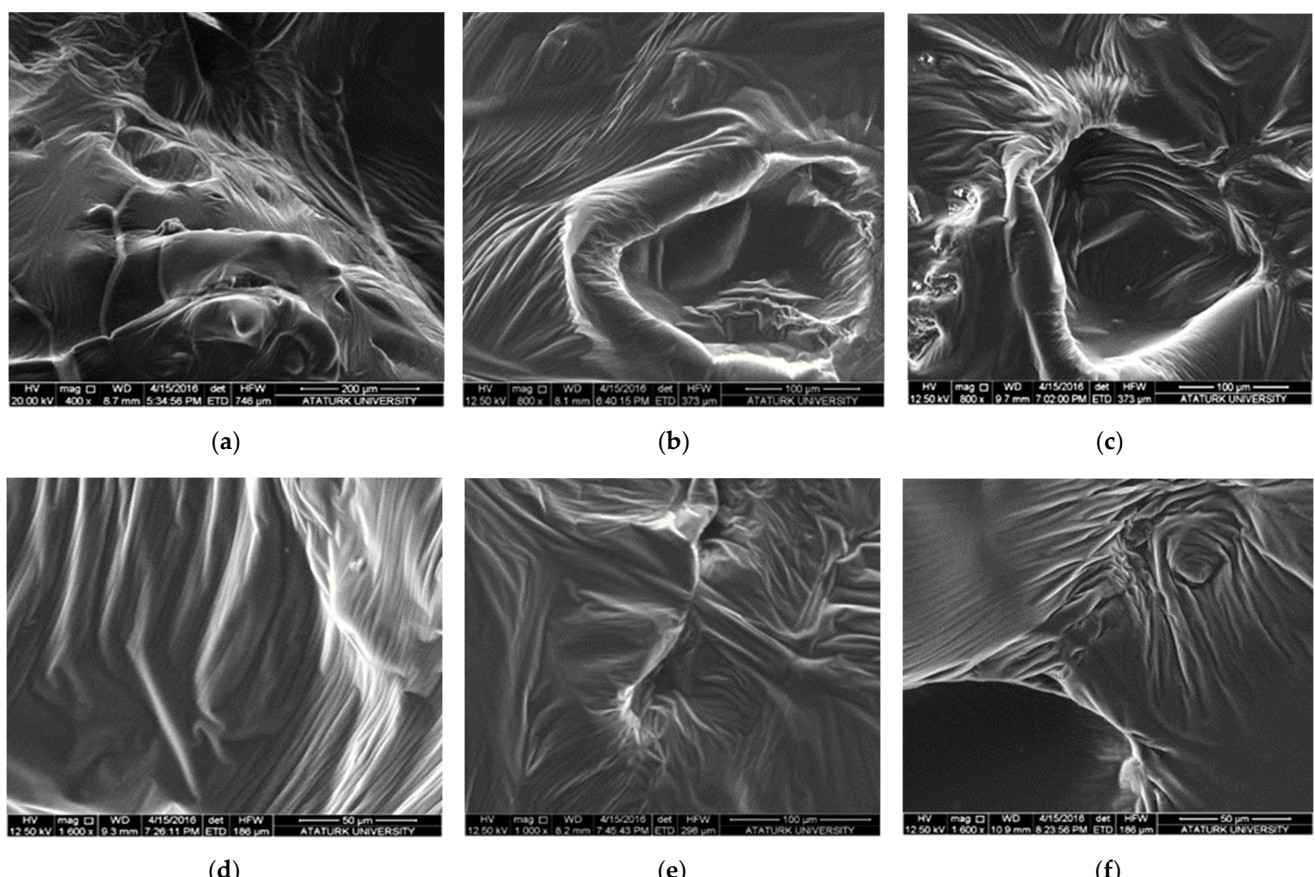

**Figure 3.** SEM images of base bitumen (B) (**a**) and mastic mixtures (CM1 (**b**), CM2 (**c**), CM3 (**d**), CM4 (**e**) and CM5 (**f**)) prepared in different filler/binder (organo-clay/bitumen) ratios.

From Figure 3a, it can be seen that the base bitumen has relatively flat, low-density folds and a soft-appearing morphological structure. In the mastic mixtures prepared at low clay ratios (Figure 3b,c), a morphological structure with widespread and dense regular folds emerged, clearly reflecting the exfoliated distribution of the clay layers and the development of the larger surface area. Images of mastic mixtures prepared at higher organo-clay ratios (Figure 3d–f) showed densely folded soft morphological structures that may have reflected a fairly good dispersion of clay plates in the bitumen matrix and an increased surface area and increased effectiveness of adhesion interactions. However, relative thickening of the folds and progressively flattening zones, probably due to the partial aggregation of clay plates, also appeared.

*3.4. Empirical Rheological Tests and Results*

The empirical rheological test results of base bitumen (B) and mastic mixtures (CM1 -5) prepared in different filler/binder (organo-clay/bitumen) ratios and the results of base bitumen and various polymer-modified bitumen samples (PMB) (Tables 6 and 7) will be compared. Thus, the effect of organo-clay, which is used as a filler in mastic mixtures at different ratios, on base bitumen will be evaluated in terms of rheological properties such as penetration, softening point, flash point, elastic recovery, dynamic viscosity, DSR (Dynamic shear rheometer) (GRADE) and Fraass breaking point. It will also be determined whether organo-clay can be an alternative to polymer modifiers.

**Table 6.** Empirical rheological test results of base bitumen (B) and mastic mixtures (CM1 -5) prepared at different filler/binder (organo-clay/bitumen) ratios.

| Test | B | CM1 | CM2 | CM3 | CM4 | CM5 |
|---|---|---|---|---|---|---|
| Penetration (25 °C, 100 g, 5 min) | 54.6 | 53.4 | 55.6 | 54.3 | 57.1 | 53.6 |
| Softening Point (°C) | 49.2 | 49.3 | 49.7 | 50.0 | 50.0 | 50.7 |
| Flash Point (°C) | 342.0 | 338.0 | 336.0 | 340.0 | 336.0 | 340.0 |
| Elastic Recovery % | 0.0 | 0.0 | 0.0 | 0.0 | 0.0 | 0.0 |
| Dynamic Viscosity (Cp) | 440.0 | 492.5 | 470.0 | 497.5 | 515.0 | 537.5 |
| DSR (Dynamic Shear Rheometer) (GRADE) | 64.0 | 64.0 | 64.,0 | 64.0 | 64.0 | 64.0 |
| Fraass Breaking Point (°C) | −7.0 | −8.0 | −8.0 | −10.0 | −10.0 | −11.0 |

**Table 7.** Calculated penetration index (PI) and penetration viscosity number (PVN) values of base bitumen (B) and mastic mixtures (CM1-5) prepared at different filler/binder (organo-clay/bitumen) ratios.

| | Samples | | | | | |
|---|---|---|---|---|---|---|
| | B | CM1 | CM2 | CM3 | CM4 | CM5 |
| Penetration Index (PI) | −1.2072 | −1.2328 | −1.0366 | −1.0161 | −0.8967 | −0.8715 |
| Penetration Viscosity Number (PVN) | −0.7321 | −0.5988 | −0.6231 | −0.5682 | −0.4694 | −0.4749 |

The empirical rheological test results of base bitumen (B) and mastic mixtures (CM1-5) prepared at different filler/binder (organo-clay/bitumen) ratios are given in Table 6.

Table 7 shows the penetration index (PI) and penetration viscosity number (PVN) values calculated using the test results in Table 6 for base bitumen (B) and mastic mixtures (CM1-5) prepared at different filler/binder (organo-clay/bitumen) ratios.

Figure 4 shows the variation of the calculated penetration index (PI) and penetration viscosity number (PVN) values of the base bitumen (B) and mastic mixtures (CM1–5) prepared at different filler/binder (organo-clay/bitumen) ratios along with the filler/binder ratio.

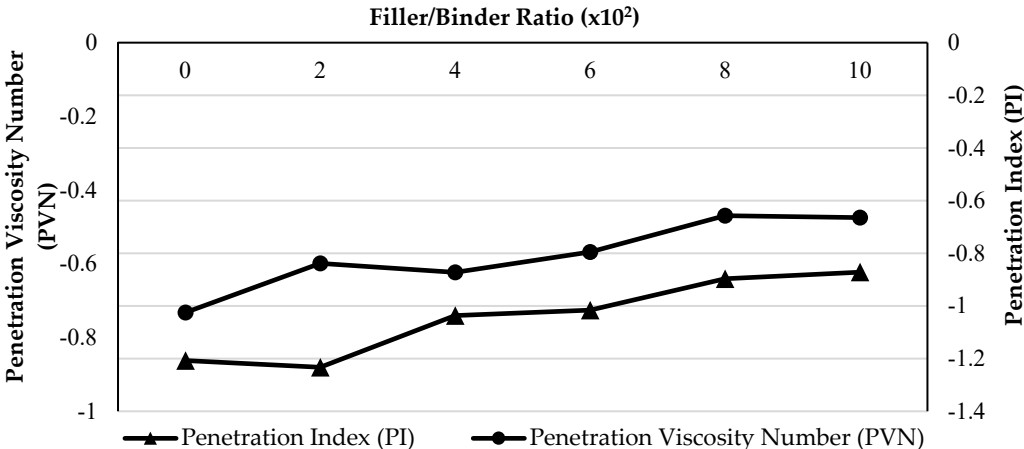

**Figure 4.** The variation of the calculated penetration index (PI) and penetration viscosity number (PVN) values of the mastic mixtures (CM1-5) prepared in different filler/binder (organoclay/bitumen) ratios, along with the filler/binder ratio and the values for the base bitumen (B).

It can be seen from Figure 4 that with an increasing organo-clay ratio, the PI values initially did not change and then tended to increase, while the PVN values increased first with a high slope and then with a relatively lower slope. On the other hand, it was observed that the softening point remained almost constant, and the dynamic viscosity increased (Table 6). The unchanged PI value indicates the homogeneity of the distribution, while the increase in the PVN value indicates hardening. The increase in PI values, in parallel with the increase in the organo-clay ratio, means that the sensitivity of mastic mixtures to temperature decreased. The increased PVN values with increasing organo-clay content can also be explained by the increased hardness of the mastic mixtures, possibly due to agglomeration of excess organo-clay layers. In addition, the PVN values calculated in all organo-clay ratios were calculated between $-1$ and $+1$ values, indicating the suitability in terms of hardness [47].

The variation of penetration values of mastic mixtures (CM1-5) prepared at different filler/binder (organo-clay/bitumen) ratios and the filler/binder ratios themselves are given in Figure 5.

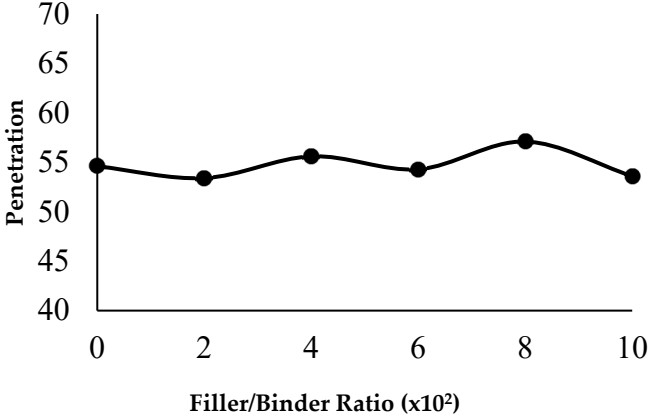

**Figure 5.** Variation of penetration values with filler/binder ratio (Base bitumen (B) and prepared mastic mixtures (CM1-5) with different filler/binder (organo-clay/bitumen) ratios).

From Figure 5, it can be seen that there is a relative increase in the penetration values in mastic mixtures with organo-clay ratios of 0.04 and 0.08, but there is a partial decrease in the mastic mixture with an organo-clay ratio of 0.10. The relative decrease observed in the highest organo-clay ratio can be explained by the relatively increased hardness due to

the partial agglomeration of the clay layers. The relative increases in penetration values compared to the base bitumen can be attributed to the possible increase in the elasticity of the bitumen due to the effective hydrophobic interactions between the organo-clay plates and the bitumen particles.

The variation of softening point values of mastic mixtures (CM1–5) prepared with different filler/binder (organo-clay/bitumen) ratios and the filler/binder ratios themselves are shown in Figure 6.

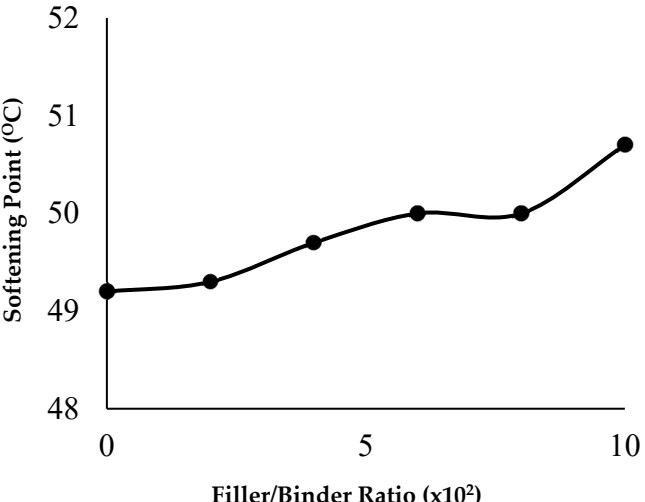

**Figure 6.** Variation of softening point values with filler/binder ratio (Base bitumen (B) and prepared mastic mixtures (CM1-5) with different filler/binder (organo-clay/bitumen) ratios).

It can be seen from Figure 6 that the softening point values increased with the increasing organo-clay ratio. This can be interpreted as a result of effective adhesion interactions between bitumen particles and lyophilic plates that are well-dispersed in the bitumen matrix and thus have an increased interaction surface. In addition, it can be evaluated that the dispersion of clay plates with a lyophilic surface does not prevent the movement of bitumen particles and does not cause textural deformation and phase separation in the bitumen matrix and even reduces shear stress. The increase in the softening point with the increase of the organo-clay ratio means that the resistance to deformation increases at high temperatures, while the relative increase in the penetration values with the increasing organo-clay ratio may means that the elasticity of the bitumen increases partially, and thus, the possibility of cracking and fracture at low temperatures will decrease.

The variation of the Fraass breaking point values of the mastic mixtures (CM1-5) prepared with different filler/binder (organo-clay/bitumen) ratios and the filler/binder ratios themselves are shown in Figure 7.

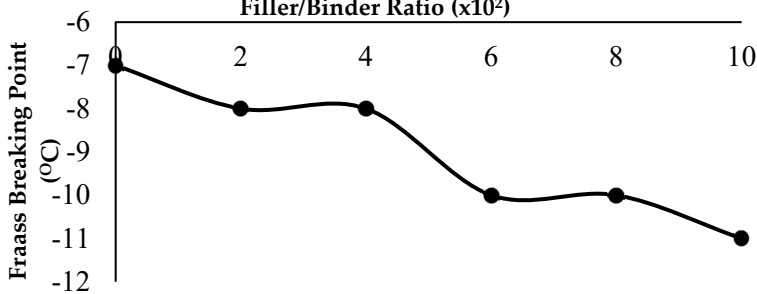

**Figure 7.** Variation of Fraass breaking point values with filler/binder ratio (Base bitumen (B) and prepared mastic mixtures (CM 1–5) with different filler/binder (organo-clay/bitumen) ratios).

It can be seen from Figure 7 that the Fraass breaking point values decreased quite regularly with the increasing organo-clay content. The decrease in the Fraass breaking point values, which is an indicator of the binder's ability to prevent cracking at low temperatures, indicated a better crack resistance at low temperature. This can be explained by the stable bulk behavior resulting from organo-clay plates exfoliating and even agglomerating at high clay ratios in the bitumen matrix and the elastic behavior resulting from increased adhesion and decreased shear stress due to the lyophilic surface properties of the clay plates. It is generally believed that the Fraass breakpoint of the binder will decrease with polymer modification [51]. It has been reported that after the matrix asphalt is modified with different polymers, the Fraass breaking point values decrease significantly, and the flexibility of the modified binders increases at low temperatures. However, there are also contradictory results showing that the Fraass breaking point of the binder increases with the addition of polymers [51–53]. It has also been reported that the Fraass breaking point of a crumb-rubber-modified binder decreases significantly with an increasing crumb rubber content. Accordingly, when the results obtained for organo-clay are compared with the polymer modification, considering the Fraass breaking point values, it can be claimed that the organo-clay can act as a modifier with a superior performance beyond being effective as a filler in a mastic mixture.

The variation of dynamic viscosity values of mastic mixtures (CM1-5) prepared with different filler/binder (organo-clay/bitumen) ratios and the filler/binder ratios themselves are shown in Figure 8.

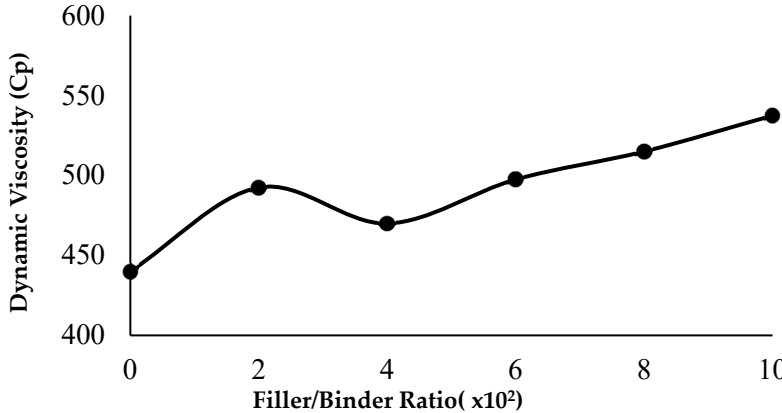

**Figure 8.** Variation of dynamic viscosity values with filler/binder ratio (Base bitumen (B) and prepared mastic mixtures (CM 1–5) with different filler/binder (organo-clay/bitumen) ratios) (135 °C).

As can be seen from Figure 8, the dynamic viscosity values of the mastic mixtures first increase with a high slope with increasing organo-clay ratio, then decrease and finally increase again with a relatively low slope. The variation of dynamic viscosity, defined as the ability of a material to resist flowing and deforming during mechanical oscillation as a function of temperature, frequency, time or both with the filler ratio is closely related to the effectiveness of interactions between organo-clay plates and bitumen components. As can be seen from the SEM images and XRD diffractogram of the mastics prepared at organo-clay ratios of 0.02 and 0.04, the clay plates were almost completely exfoliated in the bitumen matrix. Therefore, no sharp changes were observed in the flow behavior of both mastic mixtures. However, at higher organo-clay ratios, it can be said that predominantly agglomerated clay plates in the clay matrix cause a certain increase in the interaction surfaces in the bitumen bulk, which may cause relatively higher shear resistance, leading to an increase in viscosity values. In addition, the fluctuation in the penetration values of the same mastic mixtures and the sharp decrease observed in the penetration value of high clay content are parallel to the change in viscosity values. The bulk behavior of the bitumen, which causes an increase in viscosity, may have the potential to reduce the air and water permeability of the bitumen, and it is also recommended to use high viscosity

asphalt cement or modified bitumen in asphalt mixtures to reduce permeability and fill the voids [54].

The variation of dynamic shear rheometer (DSR) values of mastic mixtures (CM1-5) prepared at different filler/binder (organo-clay/bitumen) ratios and the filler/binder ratios themselves are shown in Figure 9.

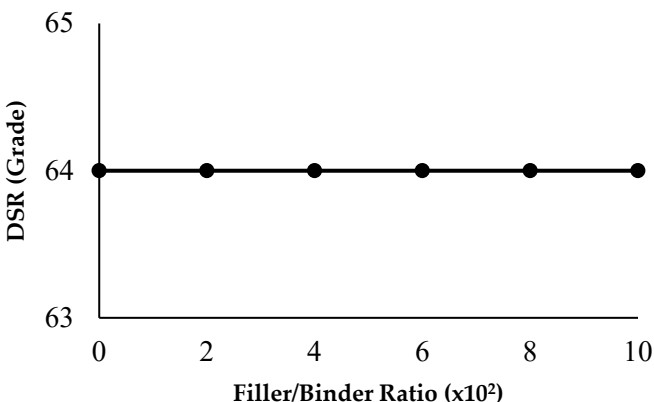

**Figure 9.** Variation of dynamic shear rheometer values (DSR) with filler/binder ratio (Base bitumen (B) and prepared mastic mixtures (CM1-5) with different filler/binder (organo-clay/bitumen) ratios).

As can be seen from Figure 9, the dynamic shear rheometer (DSR) values used to characterize the viscous and elastic behaviors of binders at medium and high temperatures did not change with an increasing organo-clay ratio, unlike the behavior of other fillers and polymer modifiers, and its value remained almost the same as that of the base bitumen. In parallel with these results, it was observed that the elastic recovery percentages did not change with the increase of the organo-clay ratio; that is, the mastics did not exhibit elastic behavior at high temperatures (Table 6).

However, the fact that DSR values did not change despite the increase in both penetration values (at organo-clay ratios of 0.04 and 0.08) and viscosity values with an increasing clay ratio suggests a different interesting effect caused by the clay layers dispersed in the bitumen matrix because while a high-penetration, less-hard mastic and increasing viscosity indicate more viscous mastic behavior, the DSR values did not change, and elasticity was not observed. These apparent contrasts can be explained by the orientations in the three-dimensional textural development of organo-clay plates dispersed in the bitumen matrix and the compatibility between the bitumen particles and the surfaces of the clay plates, which can create stable sliding or flowing behavior.

## 4. Conclusions

In this study, which focuses on the preparation and structural characterization of clay mastic, a new type of mastic material consisting of a combination of organo-clay and bituminous binder, the variation of some dynamic and rheological properties of the prepared mastic mixtures with the filler/binder (organo-clay/bitumen) ratio was investigated.

HRTEM images with XRD diffractograms of raw clay and organo-clay showed that, after lyophilization or modification, the forming plates of clay grains were widely spaced, thus making the layers available for exfoliated dispersion in the bitumen matrix. Thus, the clay plates that gained lyophobic surface properties increased their interaction potential with the bitumen particles due to the increased surface area.

It was determined that the calculated penetration index (PI) values remained constant, indicating the homogeneity of the dispersion, while the PVN values increased, indicating hardening. The increase in PI values in parallel with the increase in the organo-clay ratio means that the sensitivity of mastic mixtures to temperature decreases.

The softening point values increased with the increase of organo-clay content. This indicates increased resistance to deformation at high temperatures. On the other hand, with the increasing organo-clay ratio, the penetration values also increased relatively, which indicates that the elasticity of the bitumen increased partially, thus reducing the possibility of cracking and breaking at low temperatures.

It was found that the Fraass breaking point values decreased quite regularly with the increasing organo-clay content. The decrease in the Fraass breaking point values, which is an indicator of the binder's ability to prevent cracking at low temperatures, indicates a better crack resistance at low temperatures.

From the SEM images and XRD diffractogram of the mastics prepared at organo-clay ratios of 0.02 and 0.04, the clay plates were almost completely exfoliated within the bitumen matrix. For this reason, no sharp change was observed in the flow behavior of the mastic mixtures in both ratios. However, at higher organo-clay ratios, it was determined that the clay plates, which are predominantly agglomerated in the clay matrix, caused a certain increase in the interaction surfaces in the bitumen bulk, which caused an increase in viscosity values by causing relatively higher shear resistance.

Despite the increase in both the penetration values (at 0.04 and 0.08 organo-clay ratios) and viscosity values with increasing organo-clay ratio, the DSR values did not change. This is explained by a different and interesting effect caused by the clay layers dispersed in the bitumen matrix. While higher penetration values indicating softer mastic formation, increased viscosity and thus more viscous mastic behavior were determined, the DSR values of the mastics did not change, and elasticity was not observed. This apparent contrast is explained by the role of organo-clay plates dispersed in the bitumen matrix in three-dimensional textural development and by the adhesion interactions between the bitumen particles and the surfaces of the clay plates, which can produce harmonious sliding or flowing behavior.

**Author Contributions:** Conceptualization, A.G.; Investigation, A.G. and T.B.B.; Writing—original draft, A.G. All authors have read and agreed to the published version of the manuscript.

**Funding:** This research received no external funding.

**Data Availability Statement:** Not applicable.

**Acknowledgments:** The authors are grateful to the Istanbul Metropolitan Municipality, ISFALT Inc.to the General Directorate and the employees of bitumen laboratories for their support in the bitumen analysis in this study.

**Conflicts of Interest:** The authors declare that they have no conflict of interest.

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
