# Peer review of "Preparation, Structural Characterization and Evaluation of Some Dynamic and Rheological Properties of a New Type of Clay Containing Mastic Material, Clay-Mastic"

_minerals, doi:10.3390/min13050705_

Round 1
Reviewer 1 Report
Article "Preparation, structural characterization and evaluation of some dynamic and rheological properties of a new type of clay containing mastic material, clay-mastic" is dedicated to the actual problem of improving bitumen mastics and asphalt pavements. The authors solve this problem by using organo-modified clay, which is also an actively developing trend in recent decades. I believe that this article corresponds to the theme of the journal, but it is worth highlighting a few remarks:
1) there are a number of pronounced typos and errors in the article (caption to Fig. 9 "10-2"; there are no spaces before many references [XX]; "fraas breaking point" should be written with capital letter "Fraas breaking point"; there is no dot after [4] in line 72; "64.,0" in Table 4 for DRS value for CM2; in line 364 – it clearly refers to Fig.7, not Fig.6; and others);
2) I think the correct spelling is "organo-clay";
3) it is correct to write "Superpave";
4) decide how to write "in different" or "at different" (lines 249 and 251, respectively);
5) remove repetetive transcripts (MMT, OMMT and other) in the text;
6) the introduction lacks a description of the advantages of using montmorillonite in comparison with polymer fillers, for example, such as its environmental friendliness. Moreover, I think it is appropriate to mention the use of environmentally friendly materials for bitumen modification – both of nanoparticles (https://doi.org/10.1016/j.conbuildmat.2022.129919) and as a modifier (e.g., bio-oil https://doi.org/10.1016/j.conbuildmat.2022.127946);
7) line 146 – I think it is appropriate to refer to interesting works on the nanoparticles use for bitumen modification (e.g., https://doi.org/10.1016/j.cis.2020.102283);
8) 137 line – at least a few disadvantages of using polymers with references to literary sources needs to be written;
9) it is necessary to specify the reason for choosing this particular class of bitumen, and give more characteristics for montmorillonite (interleyered spacing and density) in the "Matherials and Methods". Also, in the same section there is absolutely no indication of the instrument and the conditions under which the rheological tests were carried out. In addition, even the temperature of viscosity measurements of the systems is absent;
10) a complete exfoliation of the MMT in the bitumen matrix should result in a substantial increase in a system viscosity. In case of this article, the viscosity increases insignificantly, indicating very little exfoliation. The SEM data cannot be evidence of a passed exfoliation. Is there an explanation for this?
11) the data of Figure 5 looks like an error. If this is not the case, please explain the wave-like behavior of the graph line. The same situation is in the case of Figure 7;
12) line 338 – can the increase in elasticity indicate not only the interaction of bitumen and filler particles, but also the interaction of clay particles among themselves? Unclear what the conclusion about the interaction of particles with bitumen is based on. What is the nature of these interactions?
13) line 349 – what was meant by bitumen particles? There are specific bitumen compounds like asphaltenes, resins or other?
14) figure 8 – there is no temperature of viscosity measurements;
15) figure 9 is uninformative. Instead of Figure 9 it is necessary to give dependence G*/sinδ on temperature, and it will be clear under which temperatures the system will be resistant to rutting;
16) line 434 – it is nesessary to reword the sentence, remove the ":", or make paragraphs in the conclusions.
Author Response
Responses to Reviewer 1
1) there are a number of pronounced typos and errors in the article (caption to Fig. 9 "10-2"; there are no spaces before many references [XX]; "fraas breaking point" should be written with capital letter "Fraas breaking point"; there is no dot after [4] in line 72; "64.,0" in Table 4 for DRS value for CM2; in line 364 – it clearly refers to Fig.7, not Fig.6; and others);
Thanks to the referee for his very useful and appropriate suggestions. All of them have been corrected.
2) I think the correct spelling is "organo-clay";
If there were two different uses, corrections were made according to the referee's suggestion.
3) it is correct to write "Superpave";
It has been corrected.
4) decide how to write "in different" or "at different" (lines 249 and 251, respectively);
It has been corrected to "at".
5) remove repetetive transcripts (MMT, OMMT and other) in the text;
It has been corrected.
6)the introduction lacks a description of the advantages of using montmorillonite in comparison with polymer fillers, for example, such as its environmental friendliness. Moreover, I think it is appropriate to mention the use of environmentally friendly materials for bitumen modification – both of nanoparticles https://doi.org/10.1016/j.conbuildmat.2022.129919) and as a modifier (e.g., bio-oil https://doi.org/10.1016/j.conbuildmat.2022.127946);
Thanks to the referee for the suggestion, the necessary additions have been made to both the text and the reference list.
7) line 146 – I think it is appropriate to refer to interesting works on the nanoparticles use for bitumen modification (e.g., https://doi.org/10.1016/j.cis.2020.102283);
Thanks to the referee for the suggestion, the necessary additions have been made to both the text and the reference list.
8) 137 line – at least a few disadvantages of using polymers with references to literary sources needs to be written;
Thanks to the referee for the suggestion, the necessary additions have been made to both the text and the reference list.
9) it is necessary to specify the reason for choosing this particular class of bitumen, and give more characteristics for montmorillonite (interleyered spacing and density) in the "Matherials and Methods". Also, in the same section there is absolutely no indication of the instrument and the conditions under which the rheological tests were carried out. In addition, even the temperature of viscosity measurements of the systems is absent;
Thanks to the referee for the suggestion, the necessary additions have been made.
10) a complete exfoliation of the MMT in the bitumen matrix should result in a substantial increase in a system viscosity. In case of this article, the viscosity increases insignificantly, indicating very little exfoliation. The SEM data cannot be evidence of a passed exfoliation. Is there an explanation for this?
The referee is right. However, the exfoliation potential of organo-clay particles in the bitumen matrix can be estimated from the images of MMT and OMMT, and the deformation of the smectite peak observed in XRD diffractogarms supports this to a certain extent. For this reason, morphological images may be considered as implying this.
11) the data of Figure 5 looks like an error. If this is not the case, please explain the wave-like behavior of the graph line. The same situation is in the case of Figure 7;
The referee is right. However, the standard deviations regarding the measurements are extremely low. In addition, a wavy appearance has emerged because the scale on the y-axis is taken as narrow in order to show the change and also because the range of change of the function is narrow depending on the variable. Perhaps if the curve were drawn as the mean, the stated situation could be eliminated without changing the trend of variation.
12) line 338 – can the increase in elasticity indicate not only the interaction of bitumen and filler particles, but also the interaction of clay particles among themselves? Unclear what the conclusion about the interaction of particles with bitumen is based on. What is the nature of these interactions?
The predominant interactions between hydrophobic clay layers and hydrophobic bitumen components are van der Waals attraction interactions and/or hydrophobic interactions.
13) line 349 – what was meant by bitumen particles? There are specific bitumen compounds like asphaltenes, resins or other?
The term bitumen particles are used to mean a complex colloidal community composed of asphaltenes, saturated paraffins, aromatic oils and resins.
14) figure 8 – there is no temperature of viscosity measurements;
It was added as 1350C
15) figure 9 is uninformative. Instead of Figure 9 it is necessary to give dependence G*/sinδ on temperature, and it will be clear under which temperatures the system will be resistant to rutting;
The referee is right. However, unfortunately, DSR measurements are limited to the failure temperature values indicating the class of bitumen.
16) line 434 – it is nesessary to reword the sentence, remove the ":", or make paragraphs in the conclusions.
That sentence has been removed
Reviewer 2 Report
The paper cannot be published in the format presented, it needs better interpretation and further characterization of the different materials. Transmission and scanning electron microscopy as well as X-ray diffraction needs a thorough review. As well as the writing of the paper is very poor and repetitive. I send the manuscript with some annotations and recommendations

Author Response
Responses to Reviewer 2
The title must be in lower case letters.
It was changed as “Preparation, structural characterization and evaluation of some dynamic and rheological properties of a new type of clay containing mastic material, clay-mastic”
Why exactly did you choose this Montmorillonite? Have you tried other montmorillonites?
Because the sample from the Çankırı basin is the most abundant of the montmorillonites in Turkey with the least impurities.
Indicating which properties influence the mechanical properties of asphalt mixtures
The physical and chemical properties, ingredients and proportions of the input materials have a great influence on the mechanical properties of asphalt mixtures used in road construction [7-23].
It was corrected as110oC
The referee may be right, but it may be difficult and not clear enough to describe the samples in the text.
- Where the microscopy analyzes were performed, laboratory data.
- The pattern must be taken from 2-65º.
- If the sample is in powder unoriented or oriented aggregated
- the spacing of each of the peaks, in figure 2
- it is also necessary to make oriented aggregates with different behaviors (EG and 350ºC heating), to compare the MMT and OMMT
Microscope images were taken in Atatürk University central laboratories.
XRD patterns were taken at the selected range for montmorillonite using powder samples to monitor the change of the layered structure based on the deformation of the characteristic smectite peak.
It was revised as “As can be seen from Fig 1b, in the case of organo-clay, the distance between the layers of the particles was wider than those in the raw clay, resulting in long stacks more separated from each other. Accordingly, organo-clay particles with larger interlayer distances and lyophilic interfaces due to modification have become more suitable for bitumen penetration and adhesion interactions.”
It was revised as” It can be seen from Figure 4 that with increasing organo clay ratio, the PI values initially did not change and then tended to increase, while the PVN values increased first with a high slope and then with a relatively lower slope. On the other hand, it was observed that the softening point remained almost constant and the dynamic viscosity increased (Table 4). The unchanged PI value indicates the homogeneity of the distribution, while the increase in the PVN value indicates hardening. The increase in PI values in parallel with the increase in the organo-clay ratio means that the sensitivity of mastic mixtures to temperature decreases. The increased PVN values with increasing organo-clay content can also be explained by the increased hardness of the mastic mixtures, possibly due to agglomeration of excess organo-clay layers. In addition, the PVN values calculated in all organo-clay ratios were calculated between -1 and +1 values, indicating the suitability in terms of hardness [44].
The variation of penetration values of mastic mixtures (CM1 -5) prepared at different filler/binder (organo clay/bitumen) ratios with filler/binder ratio is given in Fig 5.”

Round 2
Reviewer 1 Report
I believe this improved publication "Preparation, Structural Characterization and Evaluation of Some Dynamic and Rheological Properties of A New Type of Clay Containing Mastic Material, Clay-Mastic" can be accepted for publication in the journal "Minerals"
Author Response
Many thanks for your contribution.
Reviewer 2 Report
The authors have not changed anything I noted about the treatment of TEM and XRD results, they have only changed a few things concerning the format, but not the essential changes I suggested.
Author Response
Responses to referee 2
First of all, thanks to the referee for his contribution to the improvement of the manuscript.
Necessary revisions were made taking into account the referee's suggestions.
The authors have not changed anything I noted about the treatment of TEM and XRD results, they have only changed a few things concerning the format, but not the essential changes I suggested.
The referee may be right, but in this study, considering the bituminous mastic mixtures, the EG solvation method was not used. XRD patterns were taken for montmorillonite in the range of 2 θ: 3-30 using air-dried powder samples to monitor the change of layered structure based on the deformation of the characteristic smectite peak.
In this context, the paragraph has been revised as follows.
From Fig 2, it can be seen that the characteristic smectite peak of montmorillonite with a 2.1-layered structure gradually expands and shifts to the left in organo montmorillonite. The distance between plates, which was 1.23 nm for raw clay, increased to 1.66 nm for organo-clay. The smectite peak observed at 2 θ:6.2 in raw clay shifted to 5.2, which may indicate bitumen intercalation in the interlayer space. On the other hand, calcite, quartz and montmorillonite peaks appeared at 2 θ values corresponding to 28.9, 26.0 and 21.2, respectively, and the peak at 21.2 was deformed in both organo clay and mastic mixtures. This clearly indicates the expansion in interlayer spacing. The diffractograms of mastic mixtures prepared at different filler / binder ratios also show that the intensity of the smectite peak gradually increases with the increase in the organo-clay ratio, but the smectite peak disappears almost completely especially at low organo-clay ratios and the others are partially shifted to the left and have relatively lower peak intensity compared to organo-clay.
It has been revised as follows
As can be seen from Figure 1b, in the case of organo-clay, the clay layers appearing as aggregates of long fibers appeared with wider interlayer distances compared to raw clay. Accordingly, it can be argued that due to the modification, organo-clay particles with lyophilic interfaces and more spaced become more suitable for both bitumen penetration and adhesion interactions.

Round 3
Reviewer 2 Report
My decision remains the same, the figures and the discussion on the XRD and TEM results are not adequate, in that aspect the work needs, to spend a little more time and dedication. for example, a TEM image without measuring the distance between the layers to see that increase that you comment in the text is not useful, check some references on this. Diffractions should be done between 2 and 65º, and different treatments are necessary.
Author Response
Responses to Reviewer 2
My decision remains the same, the figures and the discussion on the XRD and TEM results are not adequate, in that aspect the work needs, to spend a little more time and dedication. for example, a TEM image without measuring the distance between the layers to see that increase that you comment in the text is not useful, check some references on this. Diffractions should be done between 2 and 65º, and different treatments are necessary.
We recognize and respect the referee's knowledge and experience, especially in clay characterization and XRD and TEM analysis. However, it would be useful and accurate to emphasize that this study focuses on bituminous mastic mixtures containing organo-clay. Therefore, our experience in other studies related to the clay used has shown that it will be sufficient to monitor the deformation of the characteristic peaks and especially the smectite peak in the selected scanning range in order to determine the effects of modification. The results for the variation of the interlayer spacing based on the smectite peak from the diffractograms of raw and organo-clay are given quantitatively. In addition, the results obtained with HRTEM images were tried to be supported qualitatively. Moreover, this approach is widely used. Although we do not agree with the insistence that the correction is requested, we would like to point out that the specified analyzes were carried out by relevant experts in research centers outside our laboratory over a long period of time.